# Glycemic Variability in Patients with Type 2 Diabetes Mellitus (T2DM): The Role of Melatonin in a Crossover, Double-Blind, Placebo-Controlled, Randomized Study

**DOI:** 10.3390/nu15163523

**Published:** 2023-08-10

**Authors:** Wagner Martorina, Almir Tavares

**Affiliations:** 1Neuroscience Program, Institute of Biological Sciences, Federal University of Minas Gerais, Belo Horizonte 31270-901, MG, Brazil; almirtav.bh@gmail.com; 2Department of Mental Health, Faculty of Medicine, Federal University of Minas Gerais, Belo Horizonte 31270-901, MG, Brazil

**Keywords:** diabetes mellitus, melatonin, glycemic variability, crossover, double-blind, placebo-controlled, trial

## Abstract

Background: Glycemic variability in patients with type 2 diabetes mellitus (T2DM) may be associated with chronic complications of the disease. Melatonin is a hormone that plays a crucial role in biological rhythms. Previous studies have indicated that individuals with T2DM often exhibit reduced melatonin production. In this study, our objective was to investigate whether nighttime melatonin supplementation could mitigate glycemic variability in these patients. Methods: Crossover, double-blind, placebo-controlled, randomized study. A total of 30 patients were enrolled in this study. The study included 15 participants who followed the intervention sequence of placebo (7 days)—washout (7 days)—melatonin (3 mg) (7 days), and another 15 participants who followed the sequence of melatonin (3 mg) (7 days)—washout (7 days)—placebo (7 days). During the final three days of the first and third weeks, the participants measured their pre- and postprandial capillary blood glucose levels. This study was reported according to the CONSORT 2010 statement: extension to randomized crossover trials. Results: There was a significant absolute difference in the breakfast blood glucose levels (*p* = 0.016) on Day 7. The use of melatonin determined a greater positive variation between pre- and postprandial glycemia than the placebo. The difference in glycemic amplitude between post-dinner Day 6 and pre-breakfast Day 7 was also significantly higher in the melatonin group (*p* = 0.032). Conclusions: Melatonin increased glycemic variability in individuals with type 2 diabetes mellitus (T2DM). These results can be attributed to the residual daytime effects of melatonin, prospective proximal effects, and damage to the prospective distal effects of exogenous melatonin. Therefore, caution should be exercised when administering melatonin supplementation to patients with T2DM, taking into consideration factors such as dosage, duration of use and genetic considerations.

## 1. Introduction

Type 2 diabetes mellitus (T2DM) is a disease characterized by hyperglycemia, which, if not properly treated, can lead to acute and chronic complications such as amaurosis, lower limb amputations and renal failure [1].

The glycemic control of patients with T2DM is assessed using the glycohemoglobin molecule (HbA1c), which reflects the average blood glucose levels over the past three months. Patients with unstable glycemic control, experiencing both hyperglycemia and hypoglycemia episodes, may have misleading HbA1c results that appear to be within the target range [2]. This glycemic variability has been associated with negative outcomes in patients with T2DM [3,4].

Recent studies have supported the hypothesis that a certain range of glycemic variation throughout the day, even with the nadir and peak of glycemia at the target, may contribute to chronic microvascular complications in T2DM, as occurs with chronic hyperglycemia [5,6]. Individuals with T2DM have increased glycemic variability compared to individuals without T2DM [7]. How to explain this condition?

Previous studies have attributed an increased risk of developing type 2 diabetes mellitus to sleep disturbances (disorders of quality, duration and sleep apnea) [8,9,10]. Furthermore, previous studies have shown that such disorders negatively impact the glycemic control of patients with diabetes mellitus [11,12,13,14]. In 2017, the American Diabetes Association recommended in its current guideline that patients with diabetes mellitus have their sleep patterns evaluated. The relationship between sleep and glycemic control has generated interest in melatonin, which is a hormone involved in sleep. Numerous studies have established a link between alterations in melatonin production and an increased risk of developing type 2 diabetes mellitus (T2DM) [15]. One study revealed that individuals with prediabetes and insulin resistance exhibit lower excretion levels of melatonin metabolites in their urine (6-sulfatoxymelatonin) [16]. Numerous studies have also investigated the relationship between polymorphisms in the gene responsible for expressing melatonin MTNR1B receptors and elevated susceptibility to developing type 2 diabetes mellitus (T2DM) and circadian disorders [17,18]. Melatonin is a hormone that signals the circadian rhythm. Rhythm is not only related to the sleep–wake cycle but also the rhythm patterns associated with hunger, satiety, increased alertness and disposition for activities during the day [19]. One hypothesis is that melatonin could be involved in the rate at which our blood glucose levels fluctuate throughout the day. Its nocturnal production would determine a diurnal schedule of the rhythm of hormones, such as insulin, and also counter-insulin hormones: cortisol, growth hormone and catecholamines [20], which can modulate glycemic variability. Multiple studies investigating the relationship between type 2 diabetes mellitus (T2DM) and melatonin have explored the potential benefits of melatonin supplementation in improving glycemic control among these patients. The role of melatonin in glycemic control, however, seems poorly understood or simply misunderstood. This statement is supported by the divergence in studies between exogenous melatonin and HbA1C levels, as well as the main role played by this hormone in the regulation of the circadian rhythm [21] and as an anti-inflammatory substance that may have beneficial actions in pancreatic beta cell dysfunction and insulin resistance [22,23]. It is possible that the role of melatonin is more associated with variability, with the rhythm of glucose, than with glycemic control.

Melatonin is a complex molecule, whose feedback is not as simple as that of other hormones. It is necessary to understand well the physiology of endogenous melatonin as well as the pharmacology of exogenous melatonin in order to establish its relationship with the diurnal rhythm of glucose levels throughout the day. Another important point is to define the concept of glycemic variability. There are several ways to assess whether a patient has abnormal variation in their blood glucose or not. These differences may simply be conceptual, based on various mathematical calculations, or technological, as there are different ways of measuring a patient’s blood glucose, some using cutting-edge technologies with high costs [24].

To evaluate the role of exogenous melatonin in the glycemic variability in patients with T2DM, we developed a crossover, double-blind, placebo-controlled and randomized study. Before detailing the design of the work, some concepts will be detailed, such as glycemic variability and characteristics of endogenous and exogenous melatonin.

## 2. Materials and Methods

This study was reported according to the CONSORT 2010 statement: extension to randomized crossover trials [25].

### 2.1. Objectives

#### 2.1.1. Main Objective

The primary objective of this study was to verify whether melatonin alters the glycemic variability of patients with T2DM.

#### 2.1.2. Specific Objectives

As a secondary outcome, we evaluated whether the disparity between postprandial (after-meal) and preprandial (before-meal) blood glucose levels of fasting, lunch and dinner, separately, could be altered by the use of melatonin. This evaluation aims to determine whether melatonin use has any impact on glucose fluctuation at specific times of the day.

In order to analyze the effect of melatonin after a greater number of days of use, we analyzed the glycemic variability obtained on the last day of measuring capillary glycemic levels.

We also conducted an analysis of the glycemic variation coefficient (standard deviation/mean glycemic value) of capillary blood glucose levels during the administration of the placebo and melatonin. We considered the condition to be stable if this value was below 36% and unstable if it exceeded this threshold. Furthermore, we evaluated whether there was a statistically significant difference in the glycemic variability coefficient obtained from each patient between the use of melatonin and the placebo.

Finally, we evaluated whether the use of melatonin altered the glycemic variability between post-dinner on Day 6 and pre-breakfast on Day 7.

### 2.2. Study Design

This crossover, prospective, double-blind, randomized, placebo-controlled study was approved by the ethics committee of the Federal University of Minas Gerais (CAAE: 31990720.7.000.5149) and registered in the Brazilian registry of clinical trials under code RBr-6wg54rb. The participants involved in this study signed an informed consent form that was prepared in accordance with the Declaration of Helsinki. The protocol for this study has been accepted for publication by the JMIR research protocols. The abstract and the preprint are already available on PubMed [26]. The study was carried out in an endocrinology clinic managed by the research physician (WM) in the city of Belo Horizonte, Brazil. Thirty patients participated in this work. These patients were randomized into 2 groups: the placebo–washout–melatonin (3 mg) group, which received the placebo for 7 days, followed by 7 days without study medication (washout period), followed by 7 days of melatonin (3 mg); or, alternatively, the melatonin (3 mg)–washout–placebo group, which received melatonin for 7 days, followed by 7 days off the study medication (washout period), followed by 7 days on the placebo. The 15 patients were assigned to one of the two possible intervention orders: either the placebo followed by the melatonin or the melatonin followed by the placebo. The remaining 15 patients were then assigned to the intervention order that had not been completed yet. The patients were instructed to use the placebo or melatonin (3 mg) at 9 p.m. On the 5th, 6th and 7th days of the first week and on the 5th, 6th and 7th days of the third week, the patients underwent fasting capillary blood glucose monitoring: 2 h after breakfast, before lunch, 2 h after lunch, before dinner and 2 h after dinner. No guidance on the timing of each meal was given to patients. Our aim was not to influence the natural habits of the participants. We know, however, that in Brazil breakfast takes place around 7:00 a.m., lunch around 12:00 p.m. and dinner around 7:00 p.m. The first day of melatonin or placebo use at night was considered Day 0.

### 2.3. Blinding

Vials containing melatonin and placebo were numbered 1 and 2, not necessarily in that order. The person responsible for assigning the numbers was the only person with knowledge of which vials contained melatonin and which contained the placebo. This person did not participate in any other phase of the study. All the other people involved in this project (patients, medical researchers, statisticians and assistants) were blind to the substances contained in each vial. Upon completion of the statistical analysis, the researchers were informed which vial contained melatonin and which contained the placebo. The melatonin and placebo capsules were manipulated in the artisanal pharmacy in Belo Horizonte, production batch 0630/0122. The melatonin was imported from wuxi cim science, China, batch CS-MEL-210615. The placebo was composed of Aerosil + starch + micronized cellulose + talc. The capsules, both for the placebo (bottle 2) and the melatonin (bottle 1), were identical in terms of color (white), smell (odorless) and shape, making it impossible to distinguish which capsule was melatonin and which was the placebo. The vials were also identical except for the numbering.

### 2.4. Randomization

The patients were randomized into two groups: (1) placebo–melatonin–washout and (2) melatonin–washout–placebo. This division was carried out according to the result of a dice throw. Those who threw a number from 1 to 3 went into group 1, and those who threw numbers from 4 to 6 went into group 2. The project researchers were aware of which patients used vial 1 or 2 in the first and third weeks, but they did not know which vial contained the placebo and which contained melatonin until the data analysis was completed.

### 2.5. Flowchart (Figure 1)

Figure 1 shows the flowchart of this study. After pre-test analyses, patients were randomized into two intervention sequences and post-test data were subsequently collected.

**Figure 1 nutrients-15-03523-f001:**
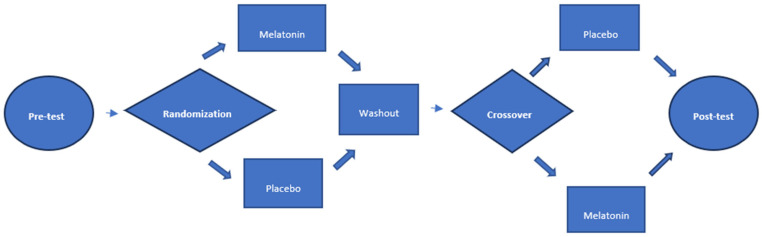
Research project flowchart.

### 2.6. Sample Size

The initial sample size was 30 patients. As referenced in the literature, when we do not know the standard deviation and population frequencies of a given variable, a pre-test should be performed with 30 to 40 patients and consider the behavior of this group as a population estimate [27]. After collecting data for the first and third weeks, we computed the standard deviation (SD) of the differences between pre- and postprandial glucose levels, both overall and for each meal, in the placebo and control groups. We established that any variation exceeding 1 SD in comparison to the placebo group would be considered significant for our calculations. We determined the required sample size (*n*) based on the different objectives of the study, and the largest *n* we found was 17 patients. If we increase the threshold for the detectable SD value, we could potentially reduce the necessary sample size even further.

### 2.7. Participants

#### 2.7.1. Inclusion Criteria

The following were included [28] in this project: (1) patients with T2DM for more than one year, (2) age greater than or equal to 40 years, (3) family history of T2DM and (4) glycohemoglobin between 7 and 10%.

#### 2.7.2. Exclusion Criteria

The following were excluded from this study: (1) pregnant women, (2) patients with recent use of corticosteroids (less than 3 months) or other hyperglycemic drugs, (3) patients who had conditions that significantly alter glycemic stability (such as in cases of renal failure, recent acute coronary syndrome and other diseases such as cancer, liver failure, etc.). Also excluded were (4) patients who refused to participate in this research at some point, and (5) those who had uncontrolled diabetes with symptoms of polydipsia, polyuria or weight loss requiring other measures for glycemic control, in addition to diet, physical activity and medications already in use. This study excluded (6) patients with high-risk criteria for severe sleep apnea, (7) depression, (8) workers who alternated work shifts and (9) patients who had epilepsy, a condition that can be aggravated by the use of melatonin.

### 2.8. Instruments

#### 2.8.1. Exogenous Melatonin

The pharmacokinetics of exogenous melatonin is influenced by several variables: dose, formulation (slow-release, immediate-release, slow-release with sustained peak), age, hepatic metabolism and time of use. In light of this, studies on the half-life, time-to-peak, and peak values have yielded conflicting results [29,30,31].

Gooneratne et al. compared the half-life of slow-release and sustained-peak melatonin at 0.4 mg and 4 mg [29]. In this work, the time-to-peak, half-life, clearance and melatonin distribution were similar in both formulations. However, melatonin values at the 4 mg dose during the peak were higher than those of endogenous melatonin. This may be a risk factor for maintaining supraphysiological doses of melatonin during the day, when this hormone, due to the action of sunlight, should reduce its concentration [29]. These results are similar to those of studies with rapid-release melatonin [31].

A better understanding of the physiology and role of endogenous melatonin will allow the development of preparations capable of mimicking this action and promoting the proper use of this substance in clinical practice. In this work, we used a dose of 3 mg of melatonin, as this is the dose most frequently used by the population.

#### 2.8.2. Assessment of Glycemic Variability

There is no gold standard for the assessment of glycemic variability [7]. All the methods for this purpose have some type of failure. The main one is to omit blood glucose variations that occur between two measurements. In this work, we included a simple methodology that was the coefficient of glycemic variability, which is the standard deviation divided by the glycemic mean. This method is validated by the literature. In addition, we calculated the glycemic variability that occurs between pre- and postprandial breakfast, lunch and dinner. The glycemic variability that occurs between close measurements is of great importance due to the speed of this oscillation. In fact, glycemia that varies rapidly between two moments is capable of triggering a series of counter-regulatory hormonal events in order to maintain homeostasis. In this sense, we believe that the inflammatory power of the variation between pre- and postprandial levels is greater than that between two preprandial glycemia (for example, pre-breakfast and pre-lunch glycemia). The calculation was performed in two ways: considering the amplitude of the difference (postprandial glucose minus the preprandial glucose). In this case, the increase or decrease in glucose by x mg/dL did not change the sign of the result, as this value was always considered positive (delta or amplitude). The other calculation method was the difference between postprandial minus the preprandial, and negative values were included in the analysis (real difference). This latter analysis seems to be significant, as prioritizing the positive or negative increment of capillary blood glucose variation is crucial. Indeed, the inflammatory potential of postprandial hyperglycemia for microvascular complications is more widely recognized than that associated with hypoglycemia. In order to evaluate the effect of melatonin on nocturnal glycemic variability, we also calculated the difference between post-dinner and pre-breakfast glycemia between Day 6 and Day 7.

#### 2.8.3. Study Variables

The clinical variables of the study were obtained through a systematic anamnesis. The following variables were included: age, sex, body mass index, insulin usage, profession, education, presence of chronic diseases, shift work status, duration since T2DM diagnosis, medications in use, use or not of insulin, use or not of sleep-inducing medications, family history of T2DM, abdominal and cervical circumference and blood pressure. HbA1C was obtained from all the patients. We limited our sample to patients with glycohemoglobin between 7 and 10%. In fact, greater glycemic variability is found in patients with a higher HbA1C value. Therefore, we did not include those with glycohemoglobin <7%. However, in order to ensure homogeneity of glycemic variability between the patients, we limited the maximum value of HbA1C to enter the study to 10%. We subsequently evaluated whether the patients who were randomized to the intervention in the melatonin (3 mg)–washout–placebo order had mean HbA1C values different from those who were randomized to the placebo–washout–melatonin (3 mg) order.

Creation of study variables

Main objective

A general variable (GV) was obtained by averaging the sum of the difference in blood glucose after breakfast (AB) and before breakfast (BB), after lunch (AL) and before lunch (BL), after dinner (AD) and before dinner (BD), on the last three days of the placebo and melatonin use in the study (Day 5, Day 6 and Day 7) and dividing this result by 3 [32].
{[(ABD5 − BBD5) + (ABD6 − BBD6) + (ABD7 − BBD7)]/3 + [(ALD5 − BLD5) + (ALD6 − BLD6) + (ALD7 − BLD7)]/3 + [(ADD5 − BDD5) + (ADD6 − BDD6) + (ADD7 − BDD7)/3]}/3

We obtained two different variables: the real difference, considering both negative and positive values, and another considering only positive values in the calculation (delta or amplitude).

Specific objectives

We analyzed separately the mean of the differences between the placebo and melatonin at breakfast, lunch and dinner on the last three days of the study.

We also created the same variables (general and specific) considering only the seventh day. The general variable was obtained as follows:[(ABD7 − BBD7) + (ALD7 – BLD7) + (ADD7 − BDD7)]/3.

The specific variable on Day 7 was obtained as follows:ABD7 − BBD7 (breakfast) and ALD7 − BLD7 (lunch) and ADD7 − BDD7 (dinner).

In addition, we calculated the mean glycemic variability coefficient for each patient when using the placebo and melatonin. The glycemic variability coefficient was calculated by dividing the standard deviation by the mean.

Finally, we calculated the difference between the post-dinner glycemia of Day 6 and the pre-breakfast glycemia of Day 7. In this case, we also calculated the delta and the real difference. This variable was obtained as follows:(ADD6 − BBD7)

We used the STOP-Bang [33] questionnaire for obstructive sleep apnea screening. Patients at high risk of obstructive sleep apnea (STOP-Bang score > 4) were excluded from the study, given the prior influence of this disease on glycemic control. The cutoff point of 4 was used in order to have more specificity in the exclusion of patients with obstructive sleep apnea.

#### 2.8.4. Statistical Analysis

Quantitative variables with Gaussian distribution had their central tendency expressed as the mean and their dispersion quantified by the standard deviation. For variables with non-normal distribution, the central tendency was described by the median and its dispersion by the 25% and 75% quartiles. The Shapiro–Wilk test was used to assess normality. Categorical variables were presented using absolute frequency and percentage.

Statistical analyzes for normal quantitative variables, before and after the intervention, were performed using the paired *t*-test. When the distribution was not normal, the Wilcoxon test was used.

Patients were compared with each other based on capillary blood glucose results obtained during the first and third weeks.

The epidemiological, clinical and laboratory data of the patients were compared between groups 1 and 2 in order to assess whether the randomization process generated similar groups from a statistical point of view.

The significance level used was 0.05, with SPSS version 20.0 being the software of choice.

## 3. Results

A total of 30 patients participated in this work. Of these, 15 underwent the intervention in the order placebo—washout—melatonin, and the other 15 melatonin—washout—placebo. Table 1 presents the epidemiological and clinical data of the patients by order of intervention. Our group consisted of patients with a mean age of over 60 years, mean time of diagnosis of 13 years and obese, as the median body mass index was >30. The patients in this study had good median glycemic control: HbA1C of around 7.3%. The 75th percentile had an HbA1C of 8.10%, indicating homogeneity in the glycemic control of these patients. These data are significant as they highlight that higher HbA1C levels correspond to increased glycemic variability among patients. There was no difference between the use or not of insulin between the two intervention sequences. These data are of great importance, as patients using insulin may have greater glycemic variability. Only nine patients in our sample used insulin. The data in this table were chosen to characterize our group of patients and also to assess whether the randomization was adequate. As the *p*-value was above 0.05 in the comparison of both intervention sequences, we can conclude that the randomization of patients was adequate.

Table 2 analyzes the real difference between the postprandial and preprandial glucose levels as described in the study variable. In this case, it is important to note that some values can be negative. Some patients may have a lower postprandial glucose level compared to their preprandial glucose level, resulting in a negative difference.

The comparison between the placebo and melatonin groups showed no significant difference in the overall mean of the three days of capillary glucose measurements (*p* = 0.805). Additionally, there was no difference in the means or medians when separately analyzing glucose levels during breakfast, lunch and dinner across the three days of measurements (*p* > 0.05). Interestingly, the *p*-values increased as the measurements moved further away from the morning period. This indicates that the positive increment of glycemic variability assessed by this methodology becomes less significant between the placebo and melatonin groups as we move away from the morning period.

The comparison of glucose levels on the seventh day, however, revealed a significant difference during breakfast (*p* = 0.016). The average glycemic variation in the melatonin group was higher than in the placebo group. Therefore, the positive increment of glycemic variability during the breakfast meal was greater with the use of melatonin compared to the use of the placebo. This difference can also be seen in Figure 2. There was no significant difference between the placebo and the melatonin groups during lunch and dinner on the third day. Similar to the overall analysis of glucose levels across the three days of measurements, we observed that the *p*-value increase moved further away from breakfast.

Table 3 presents the glycemic variability data according to the delta between the postprandial and preprandial glycemia difference. For example, a patient who had preprandial glycemia of 100 mg/dL and postprandial of 120 mg/dL, with the use of melatonin, had a variation of 20 mg/dL in capillary glycemia. If, at the time when this patient used the placebo, the variation was from 100 mg/dL in the preprandial to 80 mg/dL, that is, 20 mg/dL, we consider that there was no variation, even though the postprandial glycemia was lower. Our analyzes did not show any statistically significant difference between the pre- and postprandial measures. However, what draws attention is that *p* = 0.051 on the seventh day at breakfast, was close to statistical significance, but without the ability to confirm its presence. Between the post-dinner glycemia on the sixth day and pre-breakfast on the seventh day, there was a significantly greater range of glycemia (*p* = 0.032). Table 3 and Figure 3 illustrate this result.

Table 4 presents the results of the glycemic variation coefficient, given by the standard deviation divided by the mean. This data is well validated by the literature for the assessment of glycemic variability. We found no significant differences in this data when comparing the placebo and melatonin groups, as seen by *p* > 0.05.

We also performed an exploratory analysis of our database between the patients who used insulin and those who did not use insulin. In this analysis, it was found that in patients who did not use insulin, the glycemic variability by delta value (ignoring whether positive or negative) on the seventh day at breakfast was significantly greater when melatonin was used than when the placebo was used (*p* = 0.05). At the same time, at lunchtime, the group that used melatonin had a lower glycemic variability delta than the placebo group, even considering the average of the three days at that time (*p* = 0.049).

## 4. Discussion

This was the first randomized, double-blind, placebo-controlled, crossover study to assess the role of melatonin on glycemic variability in T2DM patients. In this work, we were able to demonstrate that melatonin of 3 mg, taken at 9:00 p.m., compared to the placebo, increased the glycemic variability in these patients on the seventh day (D7). In addition, there was a greater amplitude (delta) of glycemic variability between post-dinner and pre-breakfast glycemia between D6 and D7 in the patients who used melatonin. There was no difference in the coefficient of glycemic variability between the placebo and the melatonin. These results are different from what was initially hypothesized in the study. Melatonin is considered a hormone that regulates circadian rhythm. We expected, therefore, that its use could reduce glycemic variability in patients with type 2 diabetes and not increase it. How can we explain the results found here?

Some theories may explain the positive increase in melatonin between pre- and post-breakfast on the seventh day. The effects of exogenous melatonin expected in this study are those related to the absence of this hormone during the day. Capillary blood glucose levels were measured during the day, and melatonin was taken at night. Prospective effects of melatonin occur when its concentrations are low or undetectable. One type of prospective effect of melatonin is proximal, which occurs after this substance reduces its concentration in the morning. The nocturnal inhibitory effect of melatonin on G protein-coupled receptors ends in the morning. This deactivation of the receptor generates a rebound effect of hypersensitivity in the intracellular pathway of adenyl cyclase/cAMP/PKA/CREB [34]. The hyperactivation of the pathway could increase the concentration of insulin counter-regulatory hormones: growth hormone, catecholamines and cortisol, for example. In a patient who has a melatonin deficiency, such as those with T2DM, this is likely an upregulation of G protein-coupled receptors. The sudden increase in exogenous melatonin could potentially cause greater inhibition of this pathway during the night, leading to a rebound effect of hypersecretion of hormones even more pronounced than the physiological effect in the morning. Consequently, we would observe a greater elevation in insulin resistance. The longer half-life of exogenous melatonin could delay this rebound effect to the postprandial period, justifying greater glycemic variability as seen on the seventh day of our study during breakfast. Although there may be an increase in insulin secretion through this same mechanism, which could potentially lead to a lesser rise in postprandial glucose levels, we believe that the rebound effect on various hormones that antagonize the action of insulin would be more predominant. In addition, the possible residual effect of melatonin in the morning may have shifted the rebound effect of melatonin from fasting to post-breakfast.

Higher doses of melatonin used during the night may lead to a residual effect of this hormone beyond the nighttime period [29] Melatonin can worsen glycemic control, when its concentration is elevated during the day or meal times, by increasing insulin resistance and reducing insulin secretion [35] This could explain the greater positive variation in breakfast glycemia. In this study, a dose of 3 mg of melatonin was used, which is the most used dosage in commercial preparations of this product. However, lower doses ranging from 0.1 to 0.5 mg have been recommended when the objective is to use the medication as a chronobiotic [36] Larger doses have been used for sleep disorders such as insomnia, but without a well-defined role and with occasional and residual effects [37].

The second result of this work demonstrated an increase in the amplitude between glycemia of day 6 of post-dinner and day 7 of pre-breakfast. It can be explained by the prospective distal effects of melatonin. Melatonin acts directly to suppress the activity of the Clock/Bmal1 complex. A prolonged inhibitory effect on this complex due to a non-physiological effect of exogenous melatonin could lead to an abnormal synchronization of the biological clock, making the variation in blood glucose during the night erratic.

Another important factor that may influence the results of glycemic variability in studies on melatonin and T2DM is the melatonin receptor polymorphism MTNR1B rs10830963. Some works demonstrate that the mutation of this common receptor in patients with T2DM may predispose them to a worse glycemic response with the use of melatonin [38]. Our work was carried out with T2DM patients with a family history of this disease. These patients are at increased risk of the mutation that predisposes them to a worse glycemic response to the use of melatonin. Recent work has shown that the presence of this mutation determines a reduction in insulin secretion in the presence of melatonin in patients who have a habit of eating dinner late at night when melatonin levels are already high [35].

The variation in the glycemia delta (post minus pre-breakfast) on the seventh day of the study showed a *p*-value of 0.051, indicating a strong trend toward significance. This result explains why postprandial hyperglycemia is usually greater than preprandial and, therefore, a good part of the analyzed data was similar to those evaluated considering the negative values of the difference that were significant on the seventh day. However, data that may have interfered with this result were the inclusion of patients using insulin in the studies. A stratified analysis showed that the variation in glycemia by delta was significantly higher in patients who were not using insulin. It is possible that the power of insulin on glycemic variability was a bias for the hyperglycemic effect of melatonin in the morning, according to this analysis. In the same stratified analysis, the use of melatonin in patients who did not use insulin reduced postprandial incursions, even considering the mean glycemic variation over the three days. We believe that at lunchtime, prospective proximal effect of melatonin, which increases insulin resistance, is lower, allowing a positive effect of melatonin on the glycemic variability measured by the delta. Furthermore, the chance of having an increased serum concentration of melatonin at that time is lower, so we would not have the negative influence of the MTNR1B receptor mutation described. In this sense, we believe that the prospective distal effect of the use of melatonin becomes evident, with a better regulation of the variation in glycemia at this time. This effect is related to the direct regulation of clock genes. These conclusions, however, are merely exploratory, as the previous study design did not exclude patients using insulin.

In this work, we cannot define whether a greater positive variation in blood glucose at breakfast after 7 days of using melatonin (3 mg) was a positive or negative event. As the glycemic variability of our patients was small, a statistically significant increase in postprandial glycemia in the morning may not represent an inadequate clinical result. In fact, the physiological result is that postprandial glycemia has an elevation of at least one standard deviation. A lower elevation may represent a problem for T2DM patients, exposing them to the risk of hypoglycemia by the frequent use of medications with greater hypoglycemic power. In this sense, we can hypothesize that melatonin supplementation may have reestablished a more physiological pattern of glycemic elevation in these patients, with eventual benefit from the point of view of protection against hypoglycemia. A greater amplitude of variation of post-dinner glycemia on D6 compared to pre-breakfast on D7 can be considered a negative result of the use of melatonin. Glucose variation with an average of more than 50 mg/dL during melatonin use is considered an inadequate glucose variation.

Theoretically, exogenous melatonin could be incorporated into the therapeutic arsenal of patients with insomnia. As we age, these disorders become more prevalent, which could be associated with the decline in serum levels of this hormone as well as its physiological cyclicity [39]. Furthermore, it is worth noting that melatonin does not exhibit the typical sedative effects commonly seen with benzodiazepines [40] Despite this, this substance does not seem to work well as a hypnotic drug. Studies on melatonin and insomnia show conflicting results [41]. For this reason, we did not assess patients’ sleep habits in this work. However, we excluded patients who worked shifts, as, in this case, melatonin could have a strong influence on their sleep habits and influence blood glucose for this reason. The role of exogenous melatonin seems more related to its action as a chronobiotic [42,43] and anti-inflammatory [22]. Its use, then, has been aimed at circadian rhythm disorders caused by jet lag syndrome or for workers who alternate work shifts [44,45,46]. Previous studies demonstrate that the use of this substance for 5 days was able to resynchronize the biological rhythm, causing a phase advance of up to 2 h [47]. This data can be demonstrated in our study because the effects of melatonin on glycemic variability appeared after 7 days of use.

Our work presents some critical points. The coefficient of glycemic variability, both when the patients used the placebo and when they used melatonin, was similar and small, less than 36% in most cases. These data may be related to the fact that the patients had a glycemic average practically at the target, as verified by the general average of HbA1C, which was around 7.3%. The low glycemic variability observed in our study may have concealed a potential effect of melatonin on glucose variation. Another criticism of our study is the fact that the duration of melatonin use was only 7 days. Prolonged use may lead to different outcomes regarding glycemic variability. Finally, we did not assess the presence or absence of the MTNR1B gene mutation or the melatonin concentration of our patients. There was a simple assumption that they would have this mutation more often and would be melatonin-deficient because they have T2DM.

The positive points of this study are also relevant. The fact that it is a double-blind, placebo-controlled, randomized study gives the results of this study greater methodological validity. Furthermore, the results found in this study can be explained by the physiology of melatonin. Finally, this study was carried out in a real-life context. For example, as the aim of the study was to assess the chronobiotic effects of melatonin on glycemic variability, no guidance on dietary timing was given to patients. This implies that the results of this can therefore be used in everyday clinical practice.

## 5. Conclusions

This study concluded that 3 mg of melatonin supplementation at night increases the glycemic variability of pre- and postprandial breakfast glycemia and post-dinner glycemia in relation to fasting glycemia. Factors related to the physiology of melatonin, such as residual daytime effects of melatonin, prospective proximal effects, damage to the prospective distal effects of exogenous melatonin and patients’ genetic characteristics, may explain the results of this work. It is possible that exogenous melatonin, in smaller doses, with less risk of residual effects in the morning, has a different action on glycemic variability. Future studies, with lower doses of this hormone and for a longer period of time, are necessary to better evaluate the role of this hormone in the glycemic variability of patients with T2DM.

## Figures and Tables

**Figure 2 nutrients-15-03523-f002:**
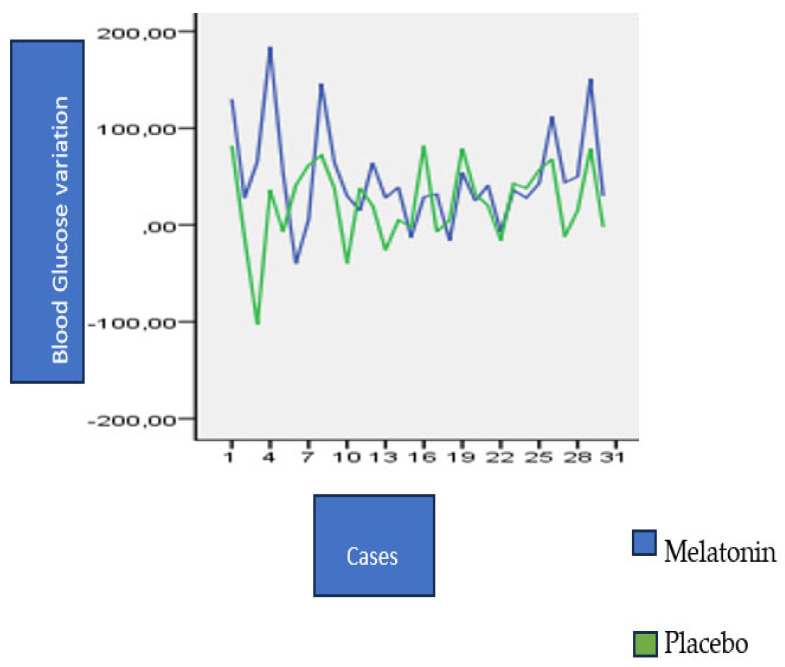
Difference between post- and preprandial breakfast glycemia on D7.

**Figure 3 nutrients-15-03523-f003:**
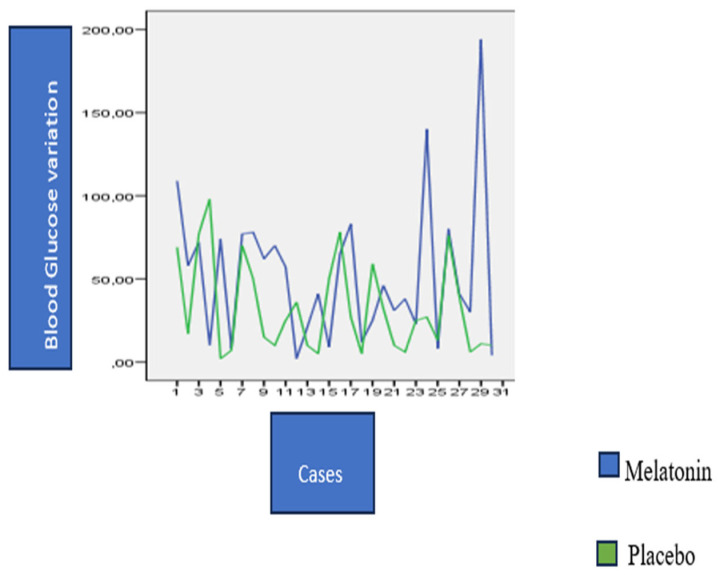
Delta or amplitude of the difference between postprandial D6 and preprandial D7.

**Table 1 nutrients-15-03523-t001:** Epidemiological data of the groups: placebo–washout–melatonin and melatonin–washout–placebo.

Variables	Melatonin → Placebo *n* = 15	Placebo → Melatonin *n* = 15	Total *n* = 30	*p-*Value
Age (mean ± SD ^5^)	62.27 ± 8.20	61.87 ± 10.95	62.07 ± 9.51	0.911 ^1^
Gender *n* (%)				
Feminine	9 (56.2%)	7 (43.8%)	16 (100.0)	
Masculine	6 (42.9%)	8 (57.1%)	14 (100.0)	0.464 ^3^
Education *n* (%)				
Elementary	0 (0.0)	3 (100.0)	3 (100.0)	
School	5 (45.5)	6 (54.4)	11 (100.0)	
faculty	10 (62.5)	6 (37.5)	16 (100.0)	0.143 ^4^
DMT2 diagnosis time				
Mean ± SD	13.20 ± 7.66	14.13 ± 7.94	13.67 ± 7.68	0.745 ^1^
Index of body mass Mean ± SD	30.06 ± 5.63	31.55 ± 6.10	30.81 ± 5.81	0.494 ^1^
Glycohemoglobin Median (Q1;Q3)	7.30 (7.10–8.10)	7.30 (7.00–8.40)	7.30 (7.10–8.10)	0.723 ^2^
Insulin use *n* (%)				
Yes	2 (22.2%)	7 (77.8%)	9 (100.0)	
No	13 (61.9%)	8 (31.1%)	21 (100.0)	0.109 ^4^

^1^ *t*-test. ^2^ Mann–Whitney test. ^3^ Asymptotic Pearson’s chi-square test. ^4^ Exact Pearson’s chi-square test. ^5^ SD standard deviation. Q1 quartile: 25%; Q3 quartile: 75%.

**Table 2 nutrients-15-03523-t002:** Analysis of the real difference between post- and preprandial.

Variation of Analyzed Glucose (Post-/Preprandial Difference)	Placebo	Melatonin	*p-*Value
General (Median (Q1–Q3))	29.16 (11.63–36.66)	25.88 (15.30–45.19)	0.805
Breakfast (mean ± standard deviation)	31.47 ± 35.30	45.5 ± 39.85	0.058
Lunch (Median (Q1–Q3))	25.83 (66.66–52.66)	22.33 (7.66–42.83)	0.57
Dinner (mean ± standard deviation)	24.34 ± 32.22	19.41 ± 36.45	0.58
Breakfast on D7 (Median (Q1–Q3))	26 (−7–58.25)	37.5 (27.25–64.25)	0.016
Lunch on D7 (mean ± standard deviation)	25.0 ± 47.14	33.13 ± 50.67	0.362
Dinner on D7 (mean ± standard deviation)	15.16 ± 40	15 ± 39.46	0.98
General on D7 (mean ± standard deviation)	20.33(8.0–42.0)	27.5 (9.0–47.5)	0.077
Night and day difference (mean ± standard deviation) D6–D7	23.7 ± 35.48	44.80 ± 50.95	0.078

Q1: quartile 25%; Q3: quartile 75%; D7: seventh day.

**Table 3 nutrients-15-03523-t003:** Analysis of the delta or amplitude of the difference between post- and preprandials.

Variation of Analyzed Glucose (Delta)	Placebo	Melatonin	*p-*Value
General (Median (Q1–Q3))	38.22 (29.94–55.75)	37.94 (29.27–56.16)	0.829
Breakfast (Median (Q1–Q3))	33.66 (21.25–67.58)	44.5 (28–68)	0.102
Lunch (Median (Q1–Q3))	42.33 (24.83–58.91)	36.5 (2.83–49.08)	0.44
Dinner (mean ± standard deviation)	37.60 ± 22.60	36.21 ± 25	0.86
Breakfast D7 (Median (Q1–Q3))	37 (12.7-63.5)	39.5 (28–64.25)	0.051
Lunch D7 (Median (Q1–Q3))	34 (20.75–71)	35 (21.75–70.5)	0.267
Dinner D7 (Median (Q1–Q3))	30.5 (10–50)	21.5 (9.5–42.25)	0.558
General D7 (Median (Q1–Q3))	37 (23.5–50.4)	37.3 (26.66–52.16)	0.29
Night and day difference D6–D7 (mean ± standard deviation)	32.16 ± 27.80	52.26 ± 42.97	0.032

D7: Seventh day; Q1: quartile 25%; Q3: quartile 75%.

**Table 4 nutrients-15-03523-t004:** Coefficient of glycemic variability (standard deviation/mean) of the placebo and melatonin groups.

Glicemic Variability	Placebo	Melatonin	*p-*Value
Glycemic variability coefficient (Median (Q1–Q3))	22.36% (18–28%)	24.52% (18.7–28%)	0.098

Q1 quartile: 25%; Q3 quartile: 75%.

## Data Availability

Data from this study will be made available by email if requested.

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
