# Peer review of "Glycemic Variability in Patients with Type 2 Diabetes Mellitus (T2DM): The Role of Melatonin in a Crossover, Double-Blind, Placebo-Controlled, Randomized Study"

_nutrients, 2023, doi:10.3390/nu15163523_

Round 1

Reviewer 1 Report

This is a well conducted study, the results are clear and their interpretation sound.

Author Response

Dear reviewer, thank you very much for carefully analyzing and understanding our work.

Wagner Martorina

Reviewer 2 Report

enticing insights to better understand the role of melatonin in patients with type 2 diabetes mellitus

Interesting results are found especially after 7 days.

The work certainly has limits due to the number which, even if not high, allows for a good statistical significance

Publication is recommended

The work is well written in correct and understandable English

Author Response

(The authors gave the same response as above.)

Reviewer 3 Report

Thank you for the opportunity to review the manuscript titled “ 

 GLYCEMIC VARIABILITY IN PATIENTS WITH TYPE 2 DIABETES MELLITUS (T2DM): THE ROLE OF MELATONIN IN A CROSSOVER, DOUBLE BLIND, PLACEBO CONTROLLED, RANDOMIZED STUDY”.

In the past decade, we have more and more research about the hormone melatonin, partly attributable to the discovery that genetic variation in MTNR1B - the melatonin receptor gene - is a risk factor for impaired fasting glucose and type 2 diabetes.

The authors conducted very interesting research that contains a lot of valuable data about glycemic variability in patients with T2DM and the role of melatonin.

The following are my comments describing these issues.

1.         In the introduction, the authors do not describe sleep habits with the level of melatonin and sleep habit and diabetes mellitus and this is very important. They focus on the complications of diabetes, but this article is not about the complications of diabetes.

In the entire article, we do not find a description of what time the patients went to bed and how many hours they slept.

Melatonin, a pineal hormone having antioxidative and anti-inflammatory properties, has been implicated in circadian-linked DM. Reduced levels of melatonin and a functional link between melatonin and insulin are implicated in the pathogenesis of type 2 diabetes.

So, what were the melatonin levels of the patients before they were included in the study?

The authors focused on a meticulous description of the study design, but we do not know whether the patients included in the study did not have sleep disorders. It is very important.

We know that nowadays, short sleep duration and sleep quality have all been linked to poor health outcomes, increasing the risk of developing metabolic diseases (T2DM). Bad sleep habits interfere with the risk of developing obesity and then diabetes, and metabolic control of both type 1 and type 2 diabetes as well.

Using the Athens Insomnia Scale would be a good solution.

2. Materials and methods Exclusion criteria

Line 176….  measures such as the use of insulin for glycemic control,

But in table 1 shows that 2 patients were taking insulin

In addition, the study group consists of people with or without obesity, so they are completely different study groups.

3. Discussion

The discussion is not a discussion, it is a repetition of the results.

There is not a single reference to other publications.

The discussion must be improved.

Author Response

Point 1: In the introduction, the authors do not describe sleep habits with the level of melatonin and sleep habit and diabetes mellitus and this is very important. They focus on the complications of diabetes, but this article is not about the complications of diabetes.

In the entire article, we do not find a description of what time the patients went to bed and how many hours they slept.

Melatonin, a pineal hormone having antioxidative and anti-inflammatory properties, has been implicated in circadian-linked DM. Reduced levels of melatonin and a functional link between melatonin and insulin are implicated in the pathogenesis of type 2 diabetes.

So, what were the melatonin levels of the patients before they were included in the study?

The authors focused on a meticulous description of the study design, but we do not know whether the patients included in the study did not have sleep disorders. It is very important.

We know that nowadays, short sleep duration and sleep quality have all been linked to poor health outcomes, increasing the risk of developing metabolic diseases (T2DM). Bad sleep habits interfere with the risk of developing obesity and then diabetes, and metabolic control of both type 1 and type 2 diabetes as well.

Using the Athens Insomnia Scale would be a good solution

Response 1: Dear reviewer, thank you very much for your contribution. In response to your requirement, we included in the article a description of the importance of sleep habits for risk and poor glycemic control in patients with type 2 diabetes mellitus. Indeed, the literature has several articles on this subject. These articles have addressed the importance of sleep and also the use of melatonin for glycemic control. In our article submitted to nutrients however we evaluated the role of melatonin in glycemic variability. We have not addressed the role of sleep and melatonin in glycemic control. This approach is new. We believe that melatonin is a drug with a more established cornobiotic power than a drug for insomnia or other sleep disorders.

In our study, we did not collect data on patients' sleep. Sleep data collection is a measure of cognitive behavioral therapy for chronic insomnia. The simple habit of writing down the times to sleep and wake up on the days of the research could change the natural sleep pattern of the patients. Our study is a real-life study and therefore may have future practical implications.

We agree that the melatonin dosage would be important before hand, but due to cost reasons this was not done. However, as the study was a crossover, the patients were compared with themselves. This may have minimized a possible bias of non-previous measurement of melatonin. We point out the lack of measurement of melatonin as a criticism of our study.

  1. Materials and methods Exclusion criteria

Line 176….  measures such as the use of insulin for glycemic control,

But in table 1 shows that 2 patients were taking insulin

In addition, the study group consists of people with or without obesity, so they are completely different study groups.

Response 2

Dear reviewer, in response to your requirement, we have improved the English of this part. Patients who used insulin did enter the study. Those who needed to start insulin or other new medications for glycemic control were included. This criterion was used to make the study ethical and so that we could only evaluate the changes that occurred with the use of placebo or melatonin and not those that occurred with the introduction of a new drug.

With regard to obesity, I would like to remind you that the patients were compared to themselves in two moments: with placebo and with melatonin. In addition, randomization ensured that the two sequences of interventions, placebo - melatonin and melatonin -placebo, had equality in terms of BMI, as shown in Table 1, where p is greater than 0.05.

  1. Discussion

The discussion is not a discussion, it is a repetition of the results.

There is not a single reference to other publications.

The discussion must be improved.

Response 3

Dear reviewer, there are no previous studies in the literature on the use of melatonin and glycemic variability. For this reason, we cannot compare our work with other studies. Previous studies on melatonin assessed whether this substance could improve glycohemoglobin levels in patients with diabetes mellitus. For this reason, most of the discussion was focused on the physiology of melatonia and how the evidence so far might explain the results. Some studies similar to this one talk about the use of melatonin during the day and the behavior of blood glucose after the postprandial test. These works were cited in the discussion, however they are different from the proposal of this work.

Finally

Dear Reviewer, To meet your requirement about references in the text, we have included articles on diabetes, sleep habits and melatonin from high impact journals. Thank you very much for the detailed observations.

Round 2

Reviewer 3 Report

The authors responded to all the reviewer's comments.

A small error crept in while correcting the manuscript

Line 54 Numerous studies has established a link between .....

I guess it should be

Numerous studies have established a link between……

Congratulations on your work.

Author Response

Dear reviewer, thank you very much for all your suggestions. We fixed the error mentioned in line 54 of the text.

Wagner Martorina